# Can Patients with Electrolyte Disturbances Be Safely and Effectively Treated in a Hospital-at-Home, Telemedicine-Controlled Environment? A Retrospective Analysis of 267 Patients

**DOI:** 10.3390/jcm13051409

**Published:** 2024-02-29

**Authors:** Cohn May, Gueron Or, Segal Gad, Zubli Daniel, Hakim Hila, Fizdel Boris, Liber Pninit, Amir Hadar, Barkai Galia

**Affiliations:** 13rd Faculty of Medicine, Charles University, 11636 Prague, Czech Republic; 2School of Medicine, University of Nicosia, Nicosia 2417, Cyprus; or.gueron@sheba.health.gov.il; 3Chaim Sheba Medical Center, Education Authority, 2nd Sheba Road, Ramat-Gan 5262000, Israel; 4Chaim Sheba Medical Center, Sheba-Beyond Virtual Hospital, 2nd Sheba Road, Ramat-Gan 5262000, Israelboris.fizdel@sheba.health.gov.il (F.B.); galia.barkai@sheba.health.gov.il (B.G.); 5Faculty of Medicine, Tel-Aviv University, Tel-Aviv 6997801, Israel; 6Chaim Sheba Medical Center, The Infection Prevention & Control Unit, 2nd Sheba Road, Ramat-Gan 5262000, Israel

**Keywords:** electrolyte disturbances, telemedicine, hospital at home, hyponatremia, hyperkalemia, hypercalcemia

## Abstract

Background. Morbidities indicated for hospital-at-home (HAH) treatment include infectious diseases and exacerbations of chronic conditions. Electrolyte disturbances are not included per se. However, their rate is high. We aimed to describe our experience via the monitoring and treatment of such patients. Methods. This was a retrospective analysis of patients in the setting of telemedicine-controlled HAH treatment. We collected data from the electronic medical records of patients who presented electrolyte disturbances. Results. For 14 months, we treated 267 patients in total in HAH settings, with a mean age of 72.2 + 16.4, 44.2% for males. In total, 261 (97.75%) patients were flagged with electrolyte disturbances, of whom 149 had true electrolyte disturbances. Furthermore, 67 cases (44.96%) had hyponatremia, 9 (6.04%) had hypernatremia after correction for hyperglycemia, 20 (13.42%) had hypokalemia and 27 (18.12%) had hyperkalemia after the exclusion of hemolytic samples. Ten (6.09%) patients had hypocalcemia and two (1.34%) had hypercalcemia corrected to albumin levels. Thirteen (8.72%) patients had hypomagnesemia and one (0.67%) had hypermagnesemia. Patients with electrolyte disturbances suffered from more chronic kidney disease (24.2% vs. 12.2%; *p* = 0.03) and malignancy (6.3% vs. 0.6%; *p* = 0.006), and were more often treated with diuretics (12.6% vs. 4.1%; *p* = 0.016). No patient died or suffered from clinically significant cardiac arrhythmias. Conclusions. The extent of electrolyte disturbances amongst HAH treatment patients is high. The monitoring and treatment of such patients can be conducted safely in this setting.

## 1. Background

### 1.1. Global Challenges Confronting Healthcare Organizations

Worldwide, there is an exacerbating challenge faced by healthcare organizations regarding both hospitalization facilities and professional workers. It is anticipated that these fundamental challenges will continue to grow, after being accelerated and accentuated by the recent years of the COVID-19 pandemic. One path for solving the aforementioned global challenge is the development of, and a diversion toward, hospital-at-home (HAH) services. Such services, which are globally advocated, could provide a viable alternative to in-hospital beds and help qualify healthcare professionals who could rise to the challenge and be recruited for these technologically enhanced routes of renovating the legacy of hospitalism [1,2]. 

### 1.2. Hospital-at-Home Alternative to In-Hospital Stay for Complicated Patients

The classic morbidities traditionally included in the realm of hospital-at-home (HAH) treatment are common infectious diseases like cellulitis, pneumonia and urinary tract infection [3,4]. In recent years, these indications have expanded to cover not only patients with infectious diseases but also those with exacerbations of chronic conditions including congestive heart failure, chronic obstructive pulmonary disease [5,6] and cancer, beyond those provided with hospice and palliative services at their homes [7]. A large body of evidence already exists that supports the fact that the hospital-at-home setting is safe, convenient and as effective as in-hospital admissions of patients with the aforementioned diagnoses [8,9,10]. Consequently, with accumulated experience, the HAH service indications will further expand [11]. Along with the expanding indications, it is only reasonable to assume that the level of severity and complexity of patients is anticipated to rise [12]. One aspect of potential patients’ complexity is their potential to present with serum electrolyte disturbances.

### 1.3. Electrolyte Disturbances as Part of HAH Service Patients’ Complexity

Electrolyte disturbances, per se, are not included in the recommended primary diagnoses for HAH services [13,14]. Nevertheless, the rate of electrolyte disturbances is high amongst patients included in the accepted HAH service indications. For example, hyponatremia frequently accompanies pneumonia [15,16], while hypercalcemia is a known complication of both benign conditions that could be part of the patient’s background, such as primary hyperparathyroidism [17] or certain malignancies that are common in the community [18]. The existing literature includes only few records for the HAH management of electrolyte disturbances: Bush and co. described a case study of a young woman who suffered from hyponatremia, associated with adrenal insufficiency (Addison crisis) due to adrenal tuberculosis, and was treated with the aid of telemedicine back in 2009 [19]. Bajaj and co. included hyponatremic patients in a cohort of hepatic encephalopathy cases they controlled via a smartphone application [20]. These and others serve only as anecdotal reports in which electrolyte disturbances were described as part of the HAH application. 

### 1.4. Our Experience with Telemedicine-Controlled HAH Services

The Sheba BEYOND hospital was established in 2020 during the first waves of the COVID-19 pandemic. Similarly to many other novel telemedicine-based services, the pandemic boosted the processes of technology’ embracement and validation in Sheba BEYOND. Our services include, amongst others, a HAH service that acts in a hybrid fashion: a nursing team reaches the patients’ home while senior physicians monitor and direct the hospitalization by applying the tools of telemedicine. Throughout the years 2020 to 2022, the milieu of our patients became more heterogeneous, with the portion of COVID-19 patients becoming smaller and the portion of non-COVID patients becoming more significant. Since we aim to produce a viable alternative to acute hospitalizations in internal medicine departments, the profile of our patients is gradually becoming more complex, with the rate of patients presenting with electrolyte disturbances becoming larger. During the past several years, alongside clinical works, we engaged in scientific questioning and the validation of practices and technologies that serve or are deemed to have the potential to be integrated into the world of telemedicine-controlled HAH services, including demonstrating the usability of a six-lead electrocardiogram for arrythmia detection by applying an ECG device that can be easily operated by the patients themselves [21]. We questioned the level of consensus amongst senior physicians regarding the interpretation of a remotely operated digital stethoscope [22] and, in a larger overview, described the future options for a model of hybrid hospital departments, assimilating both in-hospital stay with telemedicine-controlled HAH beds [23]

### 1.5. The Aim of the Current Study

The aim of this study was to describe our experience with monitoring and treating patients suffering from electrolyte disturbances in the hospital-at-home setting. 

## 2. Methods

### 2.1. Patient Data

This is a retrospective analysis of patients who remained in their homes during acute illness in the setting of telemedicine-controlled HAH. After the approval of this study by our local IRB (Institutional Review Board at the Chaim Sheba Medical Center; Approval # SMC-22-9937), we collected patients’ data from their EMRs (electronic medical records) and selected patients who suffered, at some point in time, from electrolyte disturbances during their home hospitalization. Informed consent was waived by the IRB (Institutional Review Board at the Chaim Sheba Medical Center; Approval # SMC-22-9937) due to the retrospective nature of this study. Demographic data, background morbidities, current medications, acute diagnoses, electrolyte disturbances and clinical outcomes were documented. The clinical course of patients, either improving or deteriorating, was documented. All patients included in this study were hospitalized due to classic HAH indications detailed above: urinary tract infections, pneumonia (either bacterial or viral, COVID-19 patients included) and cellulitis. Also, we included patients suffering from acute deteriorations in congestive heart failure and those with acute exacerbations of chronic obstructive pulmonary disease. 

After identifying patients that were automatically flagged, due to their EMRs, as suffering from abnormal values of blood electrolytes, we had to normalize artifacts as follows; patients who were flagged as hyponatremia were normalized using the following formulation: [Measured plasma or serum sodium concentration + (2 × (Serum glucose—100)/100)] [24]. Patients flagged as having hypocalcemia were corrected with relation to their blood albumin concentrations in accordance with the following formulation: [Calcium = Serum calcium + 0.8 × (Normal albumin—Patient albumin)] [25], where 4.0 represents the average albumin level [26]. True hyperkalemia was considered only after hemolytic samples (more than ++ in a + to ++++ score used) were excluded. The patient selection procedure and flow are described in Figure 1. 

### 2.2. Electrolyte Disturbances

Definitions of electrolyte disturbances were obtained according to severity from the up-to-date algorithms intended for the guidance of clinical patients’ management [27,28,29,30,31,32,33,34,35,36,37]. Table 1 includes the normal range of electrolytes commonly measured with definitions of severity whenever available and serving clinical purposes. 

Overall, in our HAH patient cohort, there were 149 cases (at times, more than 1 case of electrolyte disturbance appeared in the same patients) diagnosed as having electrolyte disturbances according to the definitions in Table 1 after exclusion of false values. Table 2 describes the most pathological value found in our patient population for each electrolyte disturbance (i.e., the lowest calcium concentration throughout hospitalization for a patient with hypocalcemia at any time point). 

### 2.3. Statistical Analysis

Continuous variables were expressed as mean ± standard deviation (SD) if they were normally distributed, or median with interquartile range (IQR) if they were skewed. Normality was determined using Q–Q Plots. Categorical variables were presented as frequency (%). Continuous data were compared with Student’s t-test, and categorical data were compared using chi-square or Fisher exact tests. An association was considered statistically significant for a two-sided *p* value of less than 0.05. The frequency of individual electrolyte disturbances was addressing by relating them to each case of electrolyte disturbance as an index case, even when one patient suffered from several electrolyte disturbances. Addressing the natural history and the clinical characteristics of patients with electrolyte disturbances necessitated a comparison between patients with and without electrolyte disturbances, even when some patients suffered from more than one electrolyte disturbance. All descriptive and analytical statistics were obtained using the IBM SPSS 29 statistics software. 

## 3. Results

After approval by the local ethics committee, patients’ EMRs were scanned for electrolyte concentrations. All relevant data were drawn from the electronic medical data records of 267 patients hospitalized in the Sheba-Beyond HAH service between 1 July 2021 and 1 September 2022. Extracted data included the blood levels of sodium, potassium, calcium and magnesium (taken during hospitalization for clinical reasons) along with the following patient characteristics: age, gender, height, background diseases, primary discharge diagnosis, length of hospitalization and clinical outcomes (Table 3). Patients whose electrolyte measurements were outliers (even when the same patient suffered from more than one electrolyte disturbance) of the normal range (Table 1) were separated from the others. 

The mean patient age of the whole study cohort was 72.2 + 16.4, and 44.2% of patients were males. Overall, 261 (97.75%) patients were flagged as having electrolyte disturbances at any point of time throughout their HAH stay. Overall, there were 67 cases of hyponatremia (mean sodium blood concentration: 131.07 meq/L ± 5.18); 9 cases of hypernatremia (median sodium blood concentration: 149 meq/L, IQR = 7); 20 cases of hypokalemia (mean blood potassium concentration: 3 meq/L ± 0.35); 27 cases of hyperkalemia (mean blood potassium concentration: 6.4 meq/L ± 1.17); 10 cases of hypocalcemia (mean blood calcium concentration: 8.0 mg/dL ± 0.31); 2 cases of hypercalcemia (mean blood calcium concentration: 11.69 mg/dL ± 0.27); 13 cases of hypomagnesemia (mean blood magnesium concentration: 1.5 mg/dL ± 0.22); and 1 case of hypermagnesemia (2.8 mg/dL).

Within the group of patients suffering from electrolyte disturbances, there was a significantly larger percentage of patients suffering from chronic kidney disease (serum creatinine > 1.3 mg/dL; 24.2% vs. 12.2%; *p* = 0.03), a larger portion of patients suffering from malignancy (6.3% vs. 0.6%; *p* = 0.006) and a larger portion of patients treated with diuretics (12.6% vs. 4.1%; *p* = 0.016). The following patient characteristics were not found to be significantly different between patients with and without electrolyte disturbances: BMI, gender, COVID-19 status, cardiovascular morbidities and diabetes mellitus in their background. During the aforementioned hospitalizations, none of these patients was urgently referred for in-hospital stay, died or suffered from clinically significant cardiac arrhythmias.

## 4. Discussion

Surprisingly, there are not many publications from recent years in the relevant scientific literature describing the rate of electrolyte disturbances in the general population of hospitalized patients. Ibrahim et al. described the overall rate of electrolyte disturbances in a sample of hospitalized patients in Iraq. Their findings, relating to the rate of morbidities associated with electrolyte disturbances, were somewhat different from ours; they found disturbances in 44% of admitted patients with ischemic heart diseases, 19% of patients with digestive diseases, 10.5% of patients with orthopedic surgery in an emergency (not included in our HAH population), 7% of patients with pneumonia and lung diseases, 6% of patients with diabetics, 9% of patients with sepsis and 4.5% of patients with seizures (also not included in our HAH population) [38]. Amongst hospitalized COVID-19 patients, a high frequency of electrolyte imbalances is described, with a close association between their extent of anomaly and the severity of COVID-19 infection [39]. The frequency of electrolyte disturbances in our HAH patient population was also high. Due to the fact that this domain of modern medicine is new, no previous published data could serve as a reference point for our findings in the HAH population [13]. We correlated our findings with selected patients’ background diagnoses suspected to be associated with electrolyte imbalances. 

The electrolyte disturbances amongst our patient population should not be considered surprising. We found meaningful results correlated with malignancies, chronic kidney disease and patients treated with diuretic medications. Several causes that potentially contribute to electrolyte disorders in cancer patients are anti-cancer therapies, tumor lysis syndrome and paraneoplastic syndrome of inappropriate antidiuresis and/or parathyroid-like hormonal activity. However, in most cases, we consider the cause multifactorial and frequently associated with worse patient conditions [40]. The most frequent cause for significantly higher rates of electrolyte disturbances was existing chronic kidney disease, altering a myriad of regulatory functions, resulting in both electrolyte and acid base imbalances that were potentially life-threatening [41]. 

Low serum concentrations of potassium, magnesium and calcium are potentially associated with life-threatening arrhythmias [42], while changes in sodium blood concentrations are associated, depending on their rate of development, with life-threatening neurological derangements (from seizures to coma). Magnesium is essential for the maintenance of intracellular potassium concentrations, and hypomagnesemia is often associated with hypokalemia, due to urinary potassium wasting, and hypocalcemia, due to lower parathyroid hormone secretion and end organ resistance to its effect [43]. Notwithstanding the high prevalence of electrolyte disturbances in our cohort, our patients did not experience any clinically significant cardiac arrhythmias, and no case of mortality could be attributed to these disturbances. It should be noted, however, that unless indicated by clinical circumstances, electrocardiography was conducted only upon patients’ admission.

The scientific literature is growing regarding the apparent benefits of home hospitalization, and there is a world-wide need and consensus for expanding the clinical indications for HAH services. In the future, advanced technologies are anticipated to enable the continuous telemetry of wearable devices, utilizing internet of things (IoT) for the purpose of monitoring the vital signs of patients in their homes [44,45]. Until such technologies are widely used, it is important for clinicians to make sure that home hospitalization is safe for their patients. According to the American Heart Association, one of the most feared safety concerns is the development of arrhythmias in patients (during in-hospital stays) from electrolyte disturbances [46]. In their study, Mariani et al. described a comparison of virtual visits with frontal visits in the setting of patients under cardiac electrophysiology monitoring. They found out that even in this group of patients, necessitating the utmost level of caution, there was no difference in the rate of urgent hospitalizations and that patient satisfaction was improved [47].

Our study shows that electrolyte disturbances are common in the setting of virtual medical telemedicine; nevertheless, it seems prudent enough to manage these patients at home without continuous cardiac rhythm monitoring [48]. 

During our patients’ hospitalizations, we carried out a policy of almost-daily blood tests and electrolyte blood measurements. Our results should be considered reassuring for future, further reducing the need for these tests. Such decisions, however, should rely on the results of prospective, controlled studies. 

## 5. Conclusions

According to our findings, described above, electrolyte disturbances are very common in patients hospitalized in the setting of HAH services. Since all patients included in this study complied with the worldwide consensus guidelines for HAH service indications, we conclude that our findings could be safely extrapolated to the whole population of HAH patients. Therefore, clinical guidelines addressing safety issues in the realm of home hospitalization should consider our findings valuable in relation to the high prevalence of electrolyte imbalances amongst HAH service patients. It is essential for appropriate and authorized guidelines to be introduced for blood chemistry surveillance so as to establish HAH services as a valuable replacement for the in-hospital treatment of patients. Only the meticulous follow-up of blood chemistry with early identifications of electrolyte disturbances would ensure the proper development of HAH services and reduce the future need for in-hospital beds without compromising patients’ safety. 

## 6. Limitations

This was a single-center, retrospective study. Therefore, world-wide conclusions should be drawn with precautions. The number of patients suffering from specific electrolyte disturbances was low, and some disturbances’ mean concentration values were not far from normal. Therefore, larger and both retrospective and prospective clinical trials should address the implications of electrolyte imbalances, and the terms and conditions for keeping patients with electrolyte disturbances at home. 

We did not include continuous telemonitoring in this study (since the HAH setting is applicable, in the first place, only to patients who are considered hemodynamically stable). Therefore, only clinically relevant cardiac arrhythmias were excluded. Subclinical events most probably took place but were not recorded under our surveillance. Another perspective not presented in the current study is the financial and reimbursement perspective. We recommend almost-daily blood tests for acutely ill patients at the HAH service. Also, we recommend optional daily follow-ups of ECG recordings. All these should prospectively be taken into account before designing large-scale HAH services. This study did not include data relating to the marital status and economic status of patients. Also, we did not collect enough data in order to calculate the Charleson comorbidity index. These should be taken under consideration in larger studies, with investigations of long-term clinical outcomes. 

## Figures and Tables

**Figure 1 jcm-13-01409-f001:**
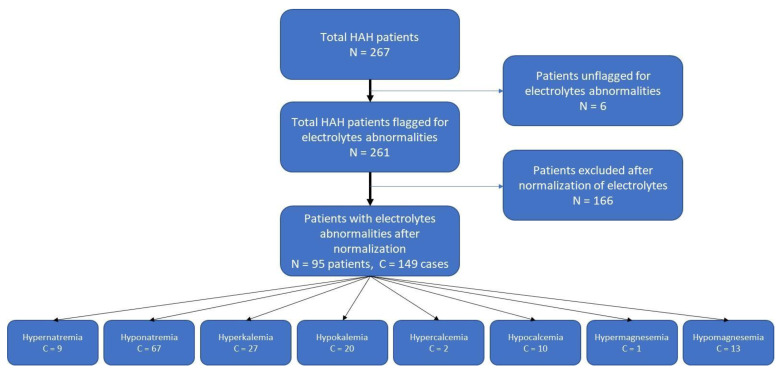
Consort flow diagram of patients.

**Table 1 jcm-13-01409-t001:** Normal range of electrolytes commonly measured with definitions of severity.

Electrolyte	Reference	Normal Range	Disturbance	Mild	Moderate	Severe
Sodium (Na)	[20,21,22]	135–145 meq/L	Hyponatremia	130–134	120–129	<120
			Hypernatremia	146–150	151–154	>155
Potassium (K)	[23,24]	3.5–5.0 meq/L	Hypokalemia	3–3.5	2.5–3	<2.5
			Hyperkalemia	5–5.5	5.6–6.4	>6.5
Calcium (Ca)	[25,26,27]	8.5–10.5 mg/dL	Hypocalcemia	8.1–8.5	7.7–8	<7.6
			Hypercalcemia	10.6–11.9	12–13.9	>14
Magnesium (Mg)	[28,29,30]	1.46–2.68 mg/dL	Hypomagnesemia	1.5–1.8	1–1.4	<1
			Hypermagnesemia	2.69–3.6	3.7–4.8	>6

**Table 2 jcm-13-01409-t002:** Electrolyte concentration values in the study cohort.

Electrolyte	Disturbance	N = 149	Values	
			Mean ± SD	Median ± IQR
Sodium (Na), meq/L	Hyponatremia	67 (44.96%)	131.07 ± 5.18	-
	Hypernatremia	9 (6.04%)	-	149 ± 7
Potassium (K), meq/L	Hypokalemia	20 (13.42%)	3 ± 0.35	-
	Hyperkalemia	27 (18.12%)	6.4 ± 1.17	-
Calcium (Ca), mg/dL	Hypocalcemia	10 (6.09%	8.0 ± 0.31	
	Hypercalcemia	2 (1.34%)	11.69 ± 0.27	
Magnesium (Mg), mg/dL	Hypomagnesemia	13 (8.72%)	1.5 ± 0.22	-
	Hypermagnesemia	1 (0.67%)	2.8	-

**Table 3 jcm-13-01409-t003:** Patient demographics according to electrolyte disturbances.

Patient Characteristic	Whole Study Population N = 267	With Electrolyte Disturbance N = 95	Without Electrolyte Disturbance N = 172	*p* Value
Demographics				
Age (years)	72.1 ± 16.4	72.1 ± 16.4	72.5 ± 16.4	0.848
BMI, Body Mass Index	25.5 ± 9.7	25.5 ± 5.8	25.5 ± 5.7	1.0
Male gender (%)	118 (44.2%)	44 (46.3%)	74 (43%)	0.75
COVID-19 Status (%)	227 (85%)	74 (77.9%)	153 (88.9%)	0.48
Background diagnoses				
Malignancy [N (%)]	7 (2.6%)	6 (6.3%)	1 (0.6%)	0.006
Cardiovascular Disease [N (%)]	97 (36.3%)	41 (43.1%)	56 (32.5%)	0.24
Diabetes [N (%)]	71 (26.59%)	33 (34.7%)	38 (22.1%)	0.092
Chronic Kidney Disease [N (%)]	44 (16.5%)	23 (24.2%)	21 (12.2%)	0.03
Medications				
Diuretics [N (%)]	19 (7.11%)	12 (12.6%)	7 (4.1%)	0.016

## Data Availability

The datasets used and/or analyzed during the current study are available from the corresponding author on reasonable request.

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
