# Peer review of "Can Patients with Electrolyte Disturbances Be Safely and Effectively Treated in a Hospital-at-Home, Telemedicine-Controlled Environment? A Retrospective Analysis of 267 Patients"

_jcm, 2024, doi:10.3390/jcm13051409_

Round 1

Reviewer 1 Report

Comments and Suggestions for Authors

Congratulations to the authors for the very interesting idea of the manuscript beczuse telemedicine already represents a major tool for physicians and its potentials have nott been completely unfolded yet; however I have many comments:

-        “Non patient died nor suffered from clinically significant cardiac arrhyth- 28
mias.” Correct English

-        Lines 36-37 need to use a better syntax here

-        Lines 38-39 need references here

-        40-44 it is a single long sentence; cut in half and use a better syntax

-        I strongly suggest you to consider this paper “Mariani MV, Pierucci N, Forleo GB, Schiavone M, Bernardini A, Gasperetti A, Mitacchione G, Mei M, Giunta G, Piro A, Chimenti C, Miraldi F, Vizza CD, Lavalle C. The Feasibility, Effectiveness and Acceptance of Virtual Visits as Compared to In-Person Visits among Clinical Electrophysiology Patients during the COVID-19 Pandemic. J Clin Med. 2023 Jan 12;12(2):620.” In order to empower the idea of the feasibility and efficacy of telemedicine in different fields of medicine.

-        In the methods you should add how you measured electrolytes concentrations at home.

-        Did you find that only diuretic drugs correlated with electrolytes imbalances? ACEI/ARB/ARNI?

Explain which statistical procedure you used for calculating differences between these two groups in covid19 because a p value of 0.48 is strange.

-         

Comments on the Quality of English Language

english is clearly the weak side of this paper; authors must correct english form and syntax

Author Response

On behalf of all authors, I thank you for your meticulous work that will, undoubtfully improve the value of the final manuscript. I uploaded a point-by-point letter regarding your specific remarks. 

Gad Segal, MD

Reviewer 2 Report

Comments and Suggestions for Authors

The author retrospectively analyzed the electrolyte disturbances of 267 patients with hospital at home (HAH), which was of clinical significance.

1 How to monitor the electrolyte of a patient who is at home, and what is the frequency of routine monitoring?

2 How to deal with electrolyte disorders in patients who was at home? What is the frequency of monitoring after the patient was treated against the electrolyte disorder? How many patients have their electrolyte disorders corrected?

3 When do patients need to be admitted to hospitals for treatment because of their electrolyte disorders?

4 The author briefly reported on the clinical outcomes of patients with electrolyte disorders, how long were they followed-up, and how long was the average time of following-up for patients with electrolyte disorders?

5 The authors provided the average values (Mean ± SD or Mean ± IQR) of ndicators for patients with electrolyte disorders, and my suggesttion is to  increase their range (which can help readers understand the severity of electrolyte abnormalities in patients). Suggest placing Table 2 in the results.

6 The resolution of Figure 1 needs to be improved.

Author Response

(The authors gave the same response as above.)

Reviewer 3 Report

Comments and Suggestions for Authors

The authors should be congratulated for their work. Telemedicine has been a field in evolution since the COVID-19 outbreak. However, more studies should assess the role of novel platforms in the remote management of patients. The results shown by the authors are interestingly new. Specifically, they recorded no major adverse event associated with Electrolyte management during a Telemedicine observation and therapy administration. Such novel papers described the role of Telemedicine nowadays. Specifically, a novel mobile app has been introduced for the screening of PSA (PMID= 38051582). Electrolyte imbalance, as well as PSA screening, are two prevalent and worthy to be addressed main arguments to which the official guidelines should be directed. Moreover, the impact of Telemedicine was widely studied in other several papers based on the SEER database (PMID= 33709970, 33230694, 33155063). These results increased the relevance of the current findings. However, the manuscript is not easily readable in the Abstract part. The numbers below 20 should be spelled out and the sentence shouldn't start with a number. 

Telemedicine has a noteworthy as well as pivotal role in the management of not-married patients and those who are limited by physical problems other than COVID (PMID= 35346115, 37254934, 32484731, 36158515, 36698806). Are data available on the marital status of these patients? And on their economic status? 

I suggest including a Charleson Comorbidity index and a subgroup analysis related to it. It will be interesting to observe that the majority of patients with a CCI >3 were protected from adverse events over the study period. Moreover, it could provide a piece of clinically meaningful information. 

Comments on the Quality of English Language

The manuscript is not easily readable in the Abstract part. The numbers below 20 should be spelled out and the sentence shouldn't start with a number. The sentences appeared not logical.

Author Response

(The authors gave the same response as above.)

Round 2

Reviewer 1 Report

Comments and Suggestions for Authors

No further comments.

Reviewer 3 Report

Comments and Suggestions for Authors

The authors properly addressed my comments.